

# School health assessment tools: a systematic review of measurement in primary schools

Maryam Kazemitabar[1], Ali Moghadamzadeh[1], Mojtaba Habibi[2,3], Rezvan Hakimzadeh[1] and Danilo Garcia[4,5,6,7,8]

[1] Department of Curriculum and Instruction, Faculty of Psychology and Educational Sciences, University of Tehran, Tehran, Iran
[2] Health Promotion Research Center, Iran University of Medical Sciences, Tehran, Iran
[3] Department of Health Psychology, School of Behavioral Sciences and Mental Health (Tehran Institute of Psychiatry), Iran University of Medical Sciences, Tehran, Iran
[4] Department of Behavioral Sciences and Learning, Linköping University, Linköping, Sweden
[5] Blekinge Center of Competence, Region Blekinge, Karlskrona, Sweden
[6] Centre for Ethics, Law and Mental Health, University of Gothenburg, Gothenburg, Sweden
[7] Department of Psychology, University of Gothenburg, Gothenburg, Sweden
[8] Network for Well–Being, Sweden

Corresponding authors
Maryam Kazemitabar,
maryam.kazemi64@ut.ac.ir,
ma.kazemi64@gmail.com
Danilo Garcia,
danilo.garcia@icloud.com

## ABSTRACT

**Background.** This systematic review aimed to investigate the psychometric properties of the school health's assessment tools in primary schools through COSMIN Risk of Bias checklist. We examined the studies that have addressed the measurement properties of school-health instruments to give a clear overview of the quality of all available tools measuring school health in primary schools. This systematic review was registered in PROPERO with the Registration ID: CRD42020158158.

**Method.** Databases of EBSCOhost, PubMed, ProQuest, Wily, PROSPERO, and OpenGrey were systematically searched without any time limitation to find all full-text English journal articles studied at least one of the COSMIN checklist measurement properties of a school-health assessment tool in primary schools. The instruments should be constructed based on a school health model. The eligible studies were assessed by COSMIN Risk of Bias checklist to report their quality of methodology for each measurement property and for the whole study by rating high, moderate or low quality.

**Results.** At the final screening just seven studies remained for review. Four studies were tool development, three of them were rated as "adequate" and the other study as "very good"; five studies examined the content validity, three of them were appraised as "very good", and the two remaining as "inadequate". All seven studies measured structural validity, three of them were evaluated as "very good", three other were scored as "adequate", and the last study as "inadequate". All the seven studies investigated the internal consistency, five of them were assessed as "very good", one was rated as "doubtful", and the last one as "inadequate". Just one study examined the cross-cultural validity and was rated as "adequate". Finally, all seven studies measured reliability, two of them were rated as "very good" and the rest five studies were appraised as "doubtful". All rating was based on COSMIN checklist criteria for quality of measurement properties assessment.

**Conclusion.** The number of studies addressing school health assessment tools was very low and therefore not sufficient. Hence, there is a serious need to investigate

the psychometric properties of the available instruments measuring school health at primary schools. Moreover, the studies included in the present systematic review did not fulfill all the criteria of the COSMIN checklist for assessing measurement properties. We suggest that future studies consider these criteria for measuring psychometric properties and developing school health assessment tools.

## BACKGROUND

Childhood consists of the golden years of everyone's life influencing many aspects of the rest of their lives among which *health* is of utmost importance. That is why the impact of childhood health status on adulthood health has been widely investigated (e.g., *Wang & Shen, 2016*; *Ballard et al., 2015*; *Péneau et al., 2017*; *Kalmakis & Chandler, 2015*; *Mehlhausen-Hassoen & Winstok, 2019*). In this context, the World Health Organization defines health as ''a state of complete physical, mental and social well-being and not merely the absence of disease or infirmity'' (*World Health Organization, 1948*) and emphasizes the necessity of sufficient healthcare during childhood (*Bravo-Sanzana, Salvo & Mieres-Chacaltana, 2015*). Since children spend a tremendous time at school, the significance of providing a healthy environment at school in all aspects seems undeniable (*Pedersen, 2019*; *Albert et al., 2019*). Moreover, research shows that healthy students learn better and they acquire better educational outcomes (*Neumann et al., 2017*; *Lindegaard Nordin, Jourdan & Simovska, 2019*; *Anderson et al., 2017*; *Scott & Karberg, 2016*).

In this regard, *Allensworth & Kolbe (1987)* have presented eight components of the Coordinated School Health Program to promote health at schools, these components are: health education (1), physical education (2), health services (3), nutrition services (4), counseling, psychological and social services (5), healthy environment (6), school-site health promotion for staff (7), and family and community involvement (8) (*Lohrmann, 2008*). Furthermore, the World Health Organization has developed the Health-Promoting Schools framework in 1996 (*World Health Organization, 1996*), which introduced three main characteristics of School Curriculum, Ethos and/or Environment, and Families and/or Communities (*Langford et al., 2015*) which is divided into six key Health-Promoting School factors of healthy school policies, school's physical environment, school's social environment, community links, action competencies for healthy living, and school health care and health promotion (*World Health Organization, 2017*). Later the Whole School, Whole Community, Whole Child model released in 2014 included 11 components of health services, health education, employee wellness, counseling, psychological and social services, nutrition environment and services, physical education and physical activity, physical environment, social emotional climate, family engagement, and community involvement

(*Lewallen et al., 2015*). This model has endeavored to link health and educational outcomes and was a combination of the Coordinated School Health model and the Association for Supervision and Curriculum Development's Whole Child approach. Furthermore, the *National Association of School Nurses (2018)* has introduced the Framework for the 21th Century School Nursing Practice, a student-centered nursing care approach that connects students, family and school community. This framework is consisted of five components including standards of practice (1), care coordination (2), leadership (3), quality improvement (4), and community/public health (5) (*National Association of School Nurses, 2016*).

As briefly reviewed here, there have been various but similar models suggesting the components or factors of school health, thus, the question is if there are instruments to measure these factors. Measuring the factors of school health is important; it helps the health specials, school nurses, school principals, teachers, parents, politicians, and other stakeholders to realize what they need to improve in order to effectively promote student health, which in turn leads to better educational outcomes (*Best, Oppewal & Travers, 2018*; *Rahman et al., 2018*) promotes public health (*Kolbe, 2019*; *Birch & Auld, 2019*), improves mental health (*Deborah, 2019*; *Redfern et al., 2019*; *Holt, 2020*), increases health equity (*Peng et al., 2019*; *González, Etow & De La Vega, 2019*), promotes general health (*Boroumandfar et al., 2015*; *Krok-Schoen et al., 2018*; *Mishra et al., 2018*), prevents diseases (*Akihiro et al., 2017*; *Park et al., 2017*; *Jihene et al., 2015*), promotes physical activities (*Kelly et al., 2019*; *Dai, 2019*), healthier nutritional choices (*Anita, 2019*; *Shrestha et al., 2019*), and increases student's safety (*Voon & Ariff, 2019*; *Chalupka & Anderko, 2019*; *Mannathoko, 2019*), among many other benefits.

In this context, measuring the psychometric properties of instruments plays an important role in research, because it shows how valid and reliable a tool is and it helps researchers to choose the best tool available for their studies. In other words, the quality of tools or instruments is directly related to their psychometric properties (*Finch, 2002*; *Roach, 2006*). The most important psychometric properties are reliability and validity (*Terwee et al., 2007*). Low reliability of a tool means it lacks of generalizability power and low validity indicates the tool is not capable of measuring the intended construct (*Mauch, Rist & Kaelin, 2017*). In other words, in order for school health instruments to be of any use for health specials, school nurses, school principals, teachers, parents, politicians, and other stakeholders, the instrument needs to have sound psychometric properties.

In sum, due to the importance of good health during childhood and the necessity to measure school health using appropriate tools with good psychometric properties, this systematic review aimed to (1) identify existing instruments that measure the factors of school health at primary schools, and (2) evaluate these measures' psychometric properties. All studies have been screened for the risk of bias and quality of measurement using the modified COnsensus-based Standards for the selection of health Measurement INstruments (COSMIN) checklist. This review will help to clarify whether the current instruments, tools, scales, and indexes measure the school health's factors.

## METHODS

### Protocol and registration

The review protocol of the study used the Preferred Reporting Items for Systematic Review and Meta-Analyses (PRISMA) guidelines. This review was registered with the International Prospective Registry of Systematic Reviews (PROSPERO) (Registration ID: CRD42020158158) before starting the research according to the PRISMA guideline recommendation (*Moher et al., 2009*).

### Eligibility criteria

The criteria for the selection of the studies included in this systematic review were that the studies: (1) developed or evaluated an instrument to measure school health; (2) the instruments should measure the school health factors; (3) evaluated at least one of the psychometric properties described in COSMIN Risk of Bias checklist for a school health instrument; (4) the population of the study should be primary school children; (5) were original research journal articles (i.e., book chapters, thesis and case studies were excluded); and (6) were in English. All articles that were not available online in full text were also excluded from the study.

### Search procedures

Two of the authors (Maryam Kazemitabar and Mojtaba Habibi) performed, independently from each other, a systematic search strategy to find articles with the specified criteria. The articles were divided into three groups of "completely related", "somewhat related", and "not related". The authors selected the articles based on the relatedness of the title and abstract of the articles, then the full texts of those placed in the "completely related" and "somewhat related" groups were studied. In case of any disagreement, the second author (Ali Moghadamzadeh) intervened until consensus was reached.

The studies were searched through the EBSCO(host), ProQuest, PubMed, and Wily databases, also a gray literature search was performed in PROSPERO, OpenGrey, and Google Scholar. No publication date was considered. Each database was queried from November 2019 using the Boolean operators (AND/OR). The terms searched were: "*school health*", "*health-promoting schools*", "*coordinated school health*", "*psychometric properties*", "*reliability*", "*validity*", "*tool*", "*scale*", "*index*", "*instrument*", "*evaluation*", and "*questionnaire*". We have also conducted hand search on the references lists of the selected studies. Table 1 shows the databases and their search algorithms applied to the search strategy.

We have sent an Email to the corresponding author of one of the articles to access the full-text of it, but we have not received any reply so it has been excluded from this Systematic Review. The search for articles was finalized by December 2019. A total of 649 studies were identified for this review.

### Study selection

From the 649 articles, we excluded all articles that did not report the psychometric properties of the instrument (cf. *Brener, Pejavara & McManus, 2011*; *Chen & Lee, 2016*;

**Table 1  Databases and their search algorithms.**

| Databases | Search algorithm | Articles founded |
|---|---|---|
| EBSCO(host) | TI (("school health") AND (scale OR index OR tool OR instrument OR questionnaire OR evaluation OR "psychometric properties" OR reliability OR validity)) AND AB (("school health") AND (scale OR index OR tool OR instrument OR questionnaire OR evaluation OR "psychometric properties" OR reliability OR validity)) | 118 |
| ProQuest | ab(("school health") AND (scale OR index OR tool OR instrument OR questionnaire OR evaluation OR "psychometric properties" OR reliability OR validity)) AND ti(("school health") AND (scale OR index OR tool OR instrument OR questionnaire OR evaluation OR "psychometric properties" OR reliability OR validity)) | 38 |
| PubMed | (((((("school health")[Title] AND (scale[Title] OR index[Title] OR tool[Title] OR instrument[Title] OR questionnaire[Title] OR evaluation[Title] OR "psychometric properties"[Title] OR reliability[Title] OR validity)[Title])) AND (("school health") AND (scale OR index OR tool OR instrument OR questionnaire OR evaluation OR "psychometric properties" OR reliability OR validity)[MeSH Terms])) AND (("school health") AND (scale OR index OR tool OR instrument OR questionnaire OR evaluation OR "psychometric properties" OR reliability OR validity)) | 503 |
| Wily | ("school health") AND (scale OR index OR tool OR instrument OR questionnaire OR evaluation OR "psychometric properties" OR reliability OR validity) | 55 |
| PROSPERO | "school health" | 65 |
| OpenGrey | ("school health") AND (scale OR index OR tool OR instrument OR questionnaire OR evaluation OR "psychometric properties" OR reliability OR validity) | 173 |
| Total | | 952 |

*Burt et al., 1996*; *Weiler & Pigg Jr, 2004*) or that did not includ middle or high school students (*Kristiansen, Holmstrom & Olofsson, 2016*; *Yun et al., 2018*; *Sagatun et al., 2019*). After this procedure, seven articles were selected as clearly relevant for inclusion in the systematic review.

## Data extraction and processing

The selected studies ($n = 7$) were assessed by the modified COSMIN checklist, which was developed to assess the methodological quality of studies on measurement properties. The original COSMIN checklist contains 10 parameters. In the modified version of the COSMIN checklist used in this systematic review, however, the parameters "*criterion validity*" and "*responsiveness*" were omitted because none of the studies reported these parameters. Thus, eight parameters were used in the present systematic review: PROM development, content validity, structural validity, internal consistency, cross-cultural
validity/measurement invariance, reliability, measurement errors, and hypotheses testing for construct validity.

Content validity refers to the degree to which the items in an instrument are an adequate reflection of the construct to be measured. Structural validity is the degree to which the scores of an instrument are an adequate reflection of the dimensionality of the construct to be measured (*Mokkink et al, 2010*). Cross-cultural validity refers to the degree to which the performance of the items on a translated or culturally adapted instrument are an adequate reflection of the performance of the items of the original version of the instrument. Internal consistency refers to the degree of interrelatedness among the items and is often assessed by Cronbach's alpha (*Prinsen et al., 2018*) and more recently by using coefficient Omega (*Nima et al., in press*). Reliability refers to the overall consistency of a measure, a measure has a high reliability that results in similar scores under consistent conditions; in other words, a test is reliable if it gives the same repeated result under the same conditions. Measurement error is the difference between a measure quantity and its true value. There are two types: systematic error and random error. The appropriate statistics for calculating the measurement error are Standard Error of Measurement, Limits of Agreement, and Smallest Detectable Change (*Prinsen et al., 2018*). Hypotheses testing for construct validity shows the extent of consistency between scores of the measure with hypotheses. It can be measured by comparing the instrument's score to other instruments or difference between relevant groups; in other words, studies that performed convergent and discriminative (divergent) validity or known-groups validity would be appraised as "very good".

The option "not applicable" applies for studies that have not investigated a specific psychometric property because it has not been applicable in that study and therefore were not included for scoring. Each parameter's check box has items to assess each specific psychometric property using a four-point scale ("*very good*", "*adequate*", "*doubtful*", and "*inadequate*"). The overall quality of each psychometric property is operationalized as the lowest rating of any standard in the box is taken (i.e., "the worst score counts" principle) (*Mokkink et al., 2018*; *Prinsen et al., 2018*; *Terwee et al., 2018*). In this study each "very good" rating received score 1 and the options "adequate", "doubtful", and "inadequate" were received score 0. That is, the maximum score for a study was 8. The final quality of any individual study was rated based on high quality for any study with 50% of the scores or more, moderate quality for 30% to 50%, and low quality for below 30% of the scores.

A customized data extraction form was developed by the first reviewer (Maryam Kazemitabar), and then characteristics data of the studies and participants were extracted; the second reviewer (Mojtaba Habibi) also extracted the data independently, the results of the two extractions were compared and in case of any differences the third reviewer (Ali Moghadamzadeh) intervened to reach consensus. The data regarding characteristics of the selected studies included: first author name, year of publication, country, sample size, time to answer the items, number of items, response scale, participant age/grade, participant sex, and instrument factors.

The instruments studied in this systematic review were: Scale for Health-Promoting Schools (SHPS), Quality of Life in School, School Health Policies and Program Study

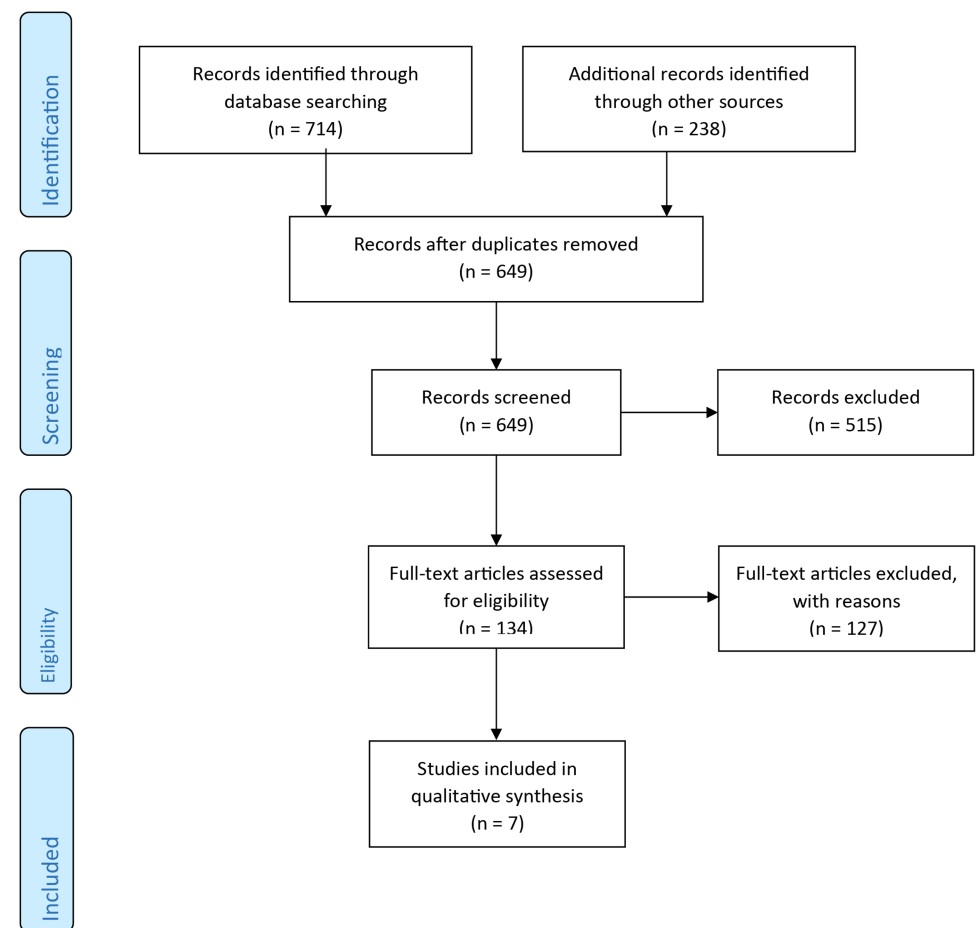

**Figure 1  PRISMA flowchart of search outcomes.**

2000 Questionnaire, Healthy School Indicator Tool (HSIT), and School Health Evaluation Instrument.

# RESULTS

## Study selection

The search strategy was conducted throughout 6 data bases: EBSCOhost, PubMed, ProQuest, Wily, PROSPERO, and OpenGrey. This resulted in 952 articles; and then in 649 after removing the duplicates. After the screening of titles and abstracts 515 articles were selected; then the full-text of the articles were studied for investigating the eligibility criteria including original journal articles, articles that studies at least one of the COSMIN Risk of Bias checklist measure properties of a school health assessment tool, study's population should be primary schools, articles of English language and articles with full-text availability. After this procedure, seven articles remained for inclusion in this systematic review (Fig. 1).

## Study characteristics

The following descriptive characteristics of the articles included in this systematic review was extracted: first author name, publication year, country, sample size, time to answer the items/number of items, response scale (multiple/forced choice, Likert scale, fill-in-the blank, true or false, yes or no, and text response), participant age/grade, participant sex, instrument's factors (Table 2).

## Risk of bias of school health measurement properties

Table 3 shows the modified COSMIN risk of bias checklist for the studies included in this systematic review. In the modified version of the checklist "criterion validity" and "responsiveness" were omitted because none of the studies reported these parameters. In this modified version, the term "tool development" substituted the term "Patient Reported Outcome Measure (PROM) development" from the original COSMIN checklist, in order to reflect the real target population of the studies which are students, teachers, and principals instead of "patient".

The articles were assessed by three raters who agreed that none of the seven articles scored "high quality", three of the articles were assessed as "moderate quality", and the remaining four articles as "low quality".

## Tool development

There are some requirements for developing an instrument. First of all the theory, framework or model used for developing the instrument should be clearly described, the target population should be specified. Context of use and qualitative data collection method should be clarified. The interviewer should be skilled and the data collected from qualitative study should be recorded and transcribed and coding section should be performed by at least two researchers independently. In addition a pilot study should be conducted to test the comprehensibility of the items, and the appropriate number of examinees for qualitative studies should be $\geq 7$ and for quantitative studies $\geq 50$ to score as "very good".

Four articles of the seven articles included in this review (*Weintraub & Erez, 2009*; *Lee et al., 2014*; *Andrews & Conte, 2005*; *Pinto et al., 2016*) were development studies; three of the studies' quality were assessed as "adequate" (*Weintraub & Erez, 2009*; *Lee et al., 2014*; *Pinto et al., 2016*) because there was not enough information whether the coding was performed independently, if the data collection continued until saturation, and if the group meetings have been recorded and transcribed. The other article (*Andrews & Conte, 2005*) was scored as "very good" because it fulfilled all criteria in the "tool development" box of the COSMIN checklist, including a clear description of steps for developing the instrument such as five focus groups indulging in determining indicators, table of specifications, creating items and response format, and conducting a pilot study. They have also mentioned that members of the advisory group have independently studied the indicators as well as reviewing relevant literature.

Kazemitabar et al. (2020), *PeerJ*, DOI 10.7717/peerj.9459

**Table 2** Characteristics of studies investigating psychometric properties of school health instruments.

| Author(s)/Publication year | Country | Sample size | Time/No. of items | Response scale | Participant age/-grade | Participant sex | Instrument factors |
|---|---|---|---|---|---|---|---|
| Öncel & Sümen (2018) | Turkey | 1236 | 15-20 min/37 | Likert | Teachers | Female & male | nutrition services, healthy school policies, physical environment, social environment, community links, individual health skills and action competencies, health services |
| Pinto et al. (2016) | Brazil | 53 | NR/28 | yes/no | School principals | Female & male | pedagogical dimension (drug and sexuality, violence and prejudice, self-care, peace and quality of life), structural dimension (access, structure conservation and equipment, sanitary conditions), relational dimension (community relations and relationship at school) |
| Ghotra et al. (2016) | Canada | 629 | NR/36 | Likert | Grades 4-6 | Female & male | teacher student relationship and social activities, physical environment, negative feelings towards school, positive feelings towards school |
| Lee et al. (2014) | Korea | 728 | NR/37 | Likert | Principals, vice principal, health teachers, physical education teachers, classroom teachers, nutritionists and school counselors | Female & male | healthy school policies, physical environment, social environment, community links, individual health skills and action competencies, health services |
| Weintraub & Erez (2009) | Israel | 526 | NR/36 | Likert | Grades 3, 4, 5, 6 | Female & male | teacher–student relationship and school activities, physical environment, negative feelings toward school, positive feelings toward school |
| Andrews & Conte (2005) | Canada | 570 | NR/87 | Likert | Teachers & Administrators | Female & male | physical health, active living, safety, nutrition, interpersonal relationships, mental health, academic health, sexual health, partnerships, substance abuse, and physical environment |
| Brener, Kann & Smith (2003) | USA | 1498 | NR/23 questionnaires | Categorical, ordinal, interval, continuous range | Elementary, middle & high school | Female & male | health services, mental health and social services, food service, school policy and environment, and faculty and staff health promotion |

**Notes.**
*NR, not reported

**Table 3  Risk of bias for school health assessment tools in primary schools.**

| Authors | Items of modified COSMIN risk of bias checklist[a] | | | | | | | | Percent | Rating |
|---|---|---|---|---|---|---|---|---|---|---|
| | 1 | 2 | 3 | 4 | 5 | 6 | 7 | 8 | | |
| Oncel & Sumen (2018) | NA | IA | VG | VG | NR | DF | NR | NR | 28% | Low quality |
| Pinto et al. (2016) | AD | VG | AD | VG | NA | VG | NR | NR | 42% | Moderate quality |
| Ghotra et al. (2016) | NA | NA | AD | VG | AD | DF | NR | VG | 33% | Moderate quality |
| Lee et al. (2014) | AD | VG | VG | VG | NA | DF | NR | NR | 42% | Moderate quality |
| Weintraub & Erez (2009) | AD | IA | VG | VG | NA | DF | NR | NR | 28% | Low quality |
| Andrews & Conte (2005) | VG | VG | AD | DF | NA | DF | NR | NR | 28% | Low quality |
| Brener, Kann & Smith (2003) | NA | NA | IA | IA | NA | VG | NR | NR | 16% | Low quality |

**Notes.**

NR, not reported; NA, not applicable; VG, very good; AD, adequate; DF, doubtful; IA, inadequate.

[a]Items of the modified COSMIN risk of bias checklist. Items: 1: tool development; 2: content validity; 3: structural validity; 4: internal consistency; 5: cross-cultural validity/measurement invariance; 6: reliability; 7: measurement errors; 8: hypotheses testing for construct validity.

## Content validity

According to the checklist the study should examine items both in an expert group and the target population to determine relevance and comprehensiveness by skilled interviewers. Other criteria are similar to the "tool development" parameter.

Five of the seven studies examined content validity of the instrument at hand. Three of them have met these criteria and received "very good" (*Lee et al., 2014*; *Andrews & Conte, 2005*; *Pinto et al., 2016*). One of the studies used the Delphi method and pilot study for evaluating content validity (*Pinto et al., 2016*); another has used focus group and a pilot study as the method of validation (*Andrews & Conte, 2005*); one study has examined only the comprehensibility of the items from students' points of view and did not evaluate content validity by asking professionals, so it was scored as "inadequate" (*Weintraub & Erez, 2009*), and the last study used only experts and did not perform a pilot study and therefore was scored "inadequate" (*Öncel & Sümen, 2018*).

## Structural validity

Measuring structural validity is only applicable for instruments that are based on a reflective model, something that all instruments included in this systematic review were. The studies that have conducted confirmatory factor analysis and had a sample size that was seven times the number of items and $\geq 100$ get scored as "very good", studies with exploratory factor analysis and a sample size at least five times the number of items and $\geq 100$ or the sample size at least six times the number of items but <100 get scored as "adequate".

All of the seven articles have investigated the structural validity of the instrument at hand. Three articles have performed confirmatory factor analysis with a sufficient sample size (seven times the number of items) and were scored as "very good" (*Öncel & Sümen, 2018*; *Weintraub & Erez, 2009*; *Lee et al., 2014*), three studies have performed exploratory factor analysis (*Ghotra et al., 2016*; *Pinto et al., 2016*; *Andrews & Conte, 2005*) and were assessed as "adequate"; the last one was assessed as "inadequate" because it has not performed confirmatory or exploratory factor analysis (*Brener, Kann & Smith, 2003*).

## Internal consistency

In COSMIN Risk of Bias checklist any study that measures internal consistency for each of the subscales independently is considered to have a "very good" quality and any study presenting only the total internal consistency or not calculating the internal consistency at all, receives the score "inadequate". Five studies have calculated Cronbach's alpha for each subscale or unidimensional scale and were therefore evaluated as "very good" (*Öncel & Sümen, 2018*; *Weintraub & Erez, 2009*; *Lee et al., 2014*; *Pinto et al., 2016*; *Ghotra et al., 2016*), one of the studies was scored as "doubtful" because internal consistency was unclear and was not fully reported (*Andrews & Conte, 2005*), the last study was scored as "inadequate" because the internal consistency has not been calculated (*Brener, Kann & Smith, 2003*).

## Cross-cultural validity/measurement invariance

Cross-cultural validity or measurement invariance is needed for a translated or culturally adapted instrument to compare performance of the items to the original version of the instrument. The samples should be similar for relevant characteristics except for the group variance; the sample size should be 200 subjects per group to be scored as "very good" and 150 subjects per group to be scored as "adequate". Cross-cultural validity assesses if the instrument is Measurement Invariance and not Differential Item Functioning occurs; this indicates that examinees from different groups with the same latent trait level act invariant to an item. Appropriate methods for Classical Test Theory (CCT) studies are regression analyses or confirmatory factor analysis, while for Item-Response Theory (IRT), the appropriate method is Differential Item Functioning (*Teresi et al., 2009*).

Only two of the studies could be investigated for cross-cultural validity. One study was scored as "adequate", because it did not report comparison between different groups clearly (*Ghotra et al., 2016*), and the other study has not reported cross-cultural validity at all (*Öncel & Sümen, 2018*).

## Reliability

As the reliability value should be stable on the construct to be measured between repeated test administrations, the time interval should be appropriate, the test condition should be similar to original instrument (e.g., administration, environment, and instructions).

Moreover, the intraclass correlation coefficient should be calculated for continuous scores. For dichotomous/nominal/ordinal scores Cohen's kappa should be calculated, for ordinal scored weighted kappa should also be calculated and it is recommended to describe weighting scheme for ordinal scores (*Prinsen et al., 2018*).

Two of the studies received score "very good" because one of them has performed split-half reliability (*Pinto et al., 2016*) and the other has performed a test-retest reliability for continuous data and Kappa has also been calculated for dichotomous scores (*Brener, Kann & Smith, 2003*). The rest five studies were scored as "doubtful" because they have not presented a clear description about stability across respondents, time interval, similar conditions or calculating the intraclass correlation coefficient.

### Measurement error

Conditions for measurement error is similar to reliability, the same examinees, appropriate time interval, and similar conditions for measurements. Standard Error of Measurement, Small Detectable Change or Limits of Agreements should be calculated. Percentage (positive and negative) agreement should be calculated for dichotomous/nominal/ordinal scores. Calculation of Standard Error Measurement through Cronbach's alpha is not appropriate, because it does not take the variance between time points into account and it should be calculated by a test-retest design (*De Vet et al., 2006*). None of the included studies calculated Standard Error of Measurement, Limits of Agreement or Smallest Detectable Change in the instruments being used in each study.

### Hypotheses testing for construct validity

Hypothesis testing for construct validity requires comparisons of the instrument under development with other outcome measurement instruments (convergent validity) through Pearson correlation and describing distribution of scores or mean scores. Performing confirmatory factor analysis or structural equation modeling over scales are also proper methods to measure differences between similar constructs. Discrepancy in scores among 'known groups' (discriminative or known group validity study) should be also measured. In discriminative or known group validity study significant characteristics of the groups under study such as age, gender, language etc. should be reported.

Only one of the studies (*Ghotra et al., 2016*) has calculated construct validity and was assessed as "very good" because it reported both convergent validity and discriminative validity for the instrument under investigation. The rest of the studies have not reported construct validity by comparing the school health instruments with another instrument or throughout known group validity.

## DISCUSSION

This systematic review assessed the measurement properties of school health assessment tools used in primary schools through the modified COSMIN methodology checklist. The selected instruments measure school health's different factors. Despite the importance of existing valid and reliable tools to measure school health, there are only a few number of school health assessment tools. Moreover, for instruments like CDC's School Health Index and CDC's School Health Profile, our search strategy was not able to find any study measuring their psychometric properties. Nevertheless, we systematically found seven studies that were recognized as eligible for our criteria. As far as we know, this is the first systematic review that has methodologically studied the quality of school health assessment tools.

The results showed that all of the seven selected studies have performed structural validity, internal consistency, and reliability. Most of the studies had satisfactory results in internal consistency, that is, they have conducted Cronbach's alpha for each subscale or have tested the instruments' unidimensionality.

Concerning structural validity and content validity, more numbers of the studies were scored "very good" among the other measurement properties after the internal consistency,

indicating a relatively strong level of evidence on these measurement properties compared to others. Conversely, none of the studies have calculated measurement error and just one study has considered the hypotheses testing for construct validity. If not specified otherwise, the random or biased errors of the scores are not obvious; which implies that we do not know the difference between the observed scores and the true scores (*Schmidt & Hunter, 1999*). That is, we do not have enough information to understand how reliable is the school health score obtained from the instrument.

Two instruments have been translated to other languages (*Öncel & Sümen, 2018*; *Ghotra et al., 2016*) and therefore should have calculated cross-cultural validity; but only one of them has assessed this property (*Ghotra et al., 2016*). Assessment of cross-cultural validity is important because the instrument may vary across different samples (e.g., healthy and patients), different gender (male and female), different languages (e.g., English and Persian), different cultures etc. (*Hjemdal et al., 2015*). Since the perception of health differs between cultures, certain school health factors might be less or more valued depending on cultural aspects (*Eves & Cheng, 2007*; *Grunert, Hieke & Juhl, 2018*; *Chiu et al., 2016*). *Öncel & Sümen (2018)* have assessed the psychometric properties of the Scale for Health Promoting Schools (*Lee et al., 2014*) in Turkish society without calculating cross-cultural validity, and *Ghotra et al. (2016)* have investigated the psychometric properties of the instrument developed by (*Weintraub & Erez, 2009*) named Quality of Life in School and estimated cross-cultural validity, although not sufficiently.

Reliability measurement requires a double test for the instrument with stable samples and proper time interval and calculating intraclass correlation coefficient or Kappa. Just two studies could be scored as "very good" (*Ghotra et al., 2016*; *Brener, Kann & Smith, 2003*) that shows the poor quality for this measurement property in the literature. Reliability is essential as it leads to trust in the instrument being applied as well as its results; for instance, if the stakeholders of school health assessment do not know how reliable the results obtained from the instrument are and whether it reproduces similar scores in different samples, then the validity of the instrument decreases as well and the results are not trustworthy (*Melchers & Beck, 2018*).

Eventually, none of the studies scored as "high quality" because they have not measured all the required measurement properties using the correct statistical methods; three studies were scored as "moderate quality" (*Pinto et al., 2016*; *Ghotra et al., 2016*; *Lee et al., 2014*); and the other four studies were scored as "low quality" (*Öncel & Sümen, 2018*; *Weintraub & Erez, 2009*; *Andrews & Conte, 2005*; *Brener, Kann & Smith, 2003*).

The Scale for Health-Promoting School' items (*Lee et al., 2014*) were developed based on the regional guideline provided by the WHO (1996) on development of Health-Promoting Schools in South Korea ($\alpha = 0.97$, RMSEA = 0.07, CFI = 0.98). The internal consistency between factors has ranged from 0.86 to 0.91. The Quality of Life at School Questionnaire (*Weintraub & Erez, 2009*) has been developed by two sources of theoretical literature related to children's quality of life in general, school quality of life and interview with students, parents, and teachers by a semi-structured questionnaire in Israel. Internal consistency among its factors has ranged from 0.68 to 0.91; correlation between total score and factors has been $0.51 < r < 0.69$. The School Health Evaluation Instrument (*Pinto et al., 2016*) has

been developed based on WHO guideline and interview by school managers of Brazil. The Kaiser-Meyer-Olkin (KMO), split-half reliability and Cronbach's alpha has been measured for each factor separately. For pedagogical dimension KMO has been 0.705, split-half test of 0.856 and Cronbach's alpha above 0.7; for structural dimension KMO has been equal to 0.639, split half reliability of 0.805 and Cronbach's alpha above 0.7; and finally for relational dimension KMO has been 0.681, split half test of 0.646 and Cronbach's alpha below 0.7.

*Brener, Kann & Smith (2003)* have measured the psychometric properties of School Health Policies and Programs Scale 2000 (SHPPS), this instrument has been developed by the Center for Disease Control and Prevention (CDC) containing 23 questionnaires. In this study Kappa has been calculated as a measure of reliability for dichotomous answers ranged from 23.3% to 88.5%, and Pearson correlation ranged from 0.38 to 0.80. This study has reported reliability for categorical, ordinal and interval items by percent without describing the exact values, but Kappa values have been reported completely. Healthy School Indicator Tool (HSIT) has been developed by *Andrews & Conte (2005)* based on Comprehensive School Health initiative (1995) and partnership of health and education professionals in Calgary, Canada. Cronbach's alpha for the nine factors have been reported above 0.7, and also a split-half reliability for all subscales and total HSIT above 0.7.

*Ghotra et al. (2016)* have measured psychometric properties of Quality of Life at School Scale in Canada. Cronbach's alpha has ranged from 0.75 to 0.93, correlation between subscales have been above 0.60. This study includes a split-half reliability but the values have not been reported. The last study performed by *Öncel & Sümen (2018)* has measured psychometric properties of the Scale for Health-Promoting Schools. They have reported Cronbach's alpha for total scale equal to 0.95 and for subscales ranged from 0.55 to 0.93, goodness of fit equal to 0.63, RMSEA of 0.12, KMO of 0.9, and a correlation between factors from 0,1 to 0.88. They have also reported a content validity index ranged from 0.91 to 1.

## Implications for school health measurement

Our findings suggest that measuring all aspects or factors of school health is not widespread. Most of the studies have measured just one, two or some factors related to school health. Measuring health at school is necessary, since it affects the policies and decisions of stakeholders of schools. Further investigations are required into measuring comprehensive and precise psychometric testing on existing tools, as well as advanced development of new tools considering appropriate statistical methods for measuring cross-cultural validity, reliability, measurement error, and construct validity. Nevertheless, among the existing tools, the Scale for Health-Promoting Schools (*Lee et al., 2014*) and the School Health Evaluation Instrument (*Pinto et al., 2016*) are the best available options for primary schools in terms of methodological evaluation. In fact, *Lee et al. (2014)* instrument has a good validity and *Pinto et al. (2016)* instrument has a good reliability. Therefore, although they require more evaluations regarding afore mentioned measurement properties, we recommend researchers, scholars and stakeholders to apply them in primary schools and to continue the psychometric testing and development of these instruments.

 

### Limitations

In sum, we found only a few instruments assessing school health. What is more, there are only a few studies measuring the psychometric properties of these instruments. Importantly, this small number of studies have not considered some of the measurement properties mentioned in the COSMIN checklist. Thus, there is not enough evidence on the quality of these instruments. This systematic review suggests that researchers need to conduct more studies on school health assessment tools in order to acquire better conclusions on these instrument's measurement properties.

## CONCLUSIONS

The studies included in this systematic review show that their instruments had a relatively strong internal consistency, a moderate quality for both structural validity and content validity, and a poor quality for tool development, reliability, cross-cultural validity, measurement error, and hypotheses testing for construct validity. Due to the inconsistency of school health assessment tools, we recommend that future studies on measurement properties of school health assessment tools adhere to the measurement properties of the COSMIN Risk of Bias checklist as it can be used as a helpful detailed manual or guideline to report the psychometric properties of school health instruments.

*"Let no one ignorant of geometry enter"*
*—Plato*

### Funding

This work was supported by a grant from The Swedish Research Council (Dnr. 2015-01229). The funders had no role in study design, data collection and analysis, decision to publish, or preparation of the manuscript.

### Grant Disclosures

The following grant information was disclosed by the authors:
The Swedish Research Council: Dnr. 2015-01229.

### Competing Interests

The authors declare there are no competing interests.

### Author Contributions

- Maryam Kazemitabar conceived and designed the experiments, performed the experiments, analyzed the data, prepared figures and/or tables, authored or reviewed drafts of the paper, and approved the final draft.
- Ali Moghadamzadeh and Mojtaba Habibi performed the experiments, analyzed the data, authored or reviewed drafts of the paper, and approved the final draft.
- Rezvan Hakimzadeh analyzed the data, prepared figures and/or tables, and approved the final draft.
- Danilo Garcia conceived and designed the experiments, analyzed the data, prepared figures and/or tables, authored or reviewed drafts of the paper, and approved the final draft.

## Data Availability

The raw data are available in the Supplementary Files.

## Supplemental Information

Supplemental information for this article can be found online at http://dx.doi.org/10.7717/peerj.9459#supplemental-information.

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
