# Peer review of "School health assessment tools: a systematic review of measurement in primary schools"

_PeerJ, doi:10.7717/peerj.9459_

## Round 0.1 · original submission · Major Revisions

Both reviewers have some critical comments demanding update the manuscript. Please compare the tools for such school students study. Show in more detail practical outcome of such research.
Please check also formatting of the table and the figure to make clear presentation. The journal readers audience is broad enough. Keep in mind to make the paper interesting for the readers.

·

Basic reporting

The manuscript is written in clear professional English.
References and background are sufficiently provided.
The manuscript adopts a professional article structure, with relevant (but improvable) results.
Results include the definitions of the different psychometrical properties (i.e. internal consistency, measurement error, etc...). I would move the definitions in the Methods section.

Experimental design

The manuscript focuses on the psychometric properties of the instruments and only superficially depicts and analyses the instruments constructs and their practical usability in the clinical or research practice.
The Authors should: 1) better justify why the psychometrical / statistical evaluation is important in the background section; 2) perform a more detailed analysis of the content of the instruments and reprise it in the discussion.

Validity of the findings

In a manuscript that evaluates instruments, the reader would expect to find implementation issues in the discussion.
The discussion section lacks of implication for the clinical or research practice. For example, what instruments should be used? Do you reccomend it? I would say that the answer to this question should include more than an evaluation of the validation methodology.

Additional comments

The third table is redundant, it could be converted in plain text. Conversely, the search algorithm (lines 143-160) could be reported in a table.

Reviewer 2 ·

Basic reporting

Thank you for the opportunity to review your manuscript! The authors address a critical issue regarding school health assessment and how the work fits into the broaden the knowledge of school health promotion. Major relevant literature has been appropriately referenced.

Other suggestions:
Title: Shouldn’t only the first alphabet of the first word be capitalized?
WHO Health-Promoting Schools Framework and WHO health-promoting components were mentioned in the article. If they present the same concept, I suggest authors to maintain consistency throughout the article.
Table 1. Please list studies reviewed in chronological order; use the correct format for Table 1 title.
Several grammar errors were identified. Please correct them throughout the manuscript. For example, line 305.
Citations in the references should follow the guideline and format provided by the journal.

Experimental design

I hope the following suggestions will help improve the quality of the study.
In the methods section, some information needs further clarifications.
Criteria for selection (2), the instruments should measure the WHO health-promoting components (at least four of the eight components). Authors introduced several models suggesting various components of school health. I wonder why the WHO health-promoting components were used? What are those eight components proposed by WHO health-promoting? I suggest authors to address them in the literature review. Also, what is the rationale of at least four of eight components should be met? Please explain. Shouldn’t those WHO health-promoting components be used as terms other than “school health” to search for additional potential articles?
Criteria for selection (4), please help me understand the rationale of the focus on primary school students? Why were middle and high schools excluded? If primary school was chosen, I suggest that authors improve the description in literature review to provide justification for your study.
The term “evaluation” was used in the search (line 151). It should be added to line 148.

Validity of the findings

If authors can clarify the issues mentioned in Experimental design section, the findings of the study will make more sense.

Results:
Please make sure that the description and numbers of articles described in the Study Selection section aligns with Figure 1.

Additional comments

Thank you for the opportunity to review your manuscript! The authors address a critical issue regarding school health assessment and how the work fits into the broaden the knowledge of school health promotion. Major relevant literature has been appropriately referenced. I hope the following suggestions will help improve the quality of the study.

In the methods section, some information needs further clarifications.
Criteria for selection (2), the instruments should measure the WHO health-promoting components (at least four of the eight components). Authors introduced several models suggesting various components of school health. I wonder why the WHO health-promoting components were used? What are those eight components proposed by WHO health-promoting? I suggest authors to address them in the literature review. Also, what is the rationale of at least four of eight components should be met? Please explain. Shouldn’t those WHO health-promoting components be used as terms other than “school health” to search for additional potential articles?
Criteria for selection (4), please help me understand the rationale of the focus on primary school students? Why were middle and high schools excluded? If primary school was chosen, I suggest that authors improve the description in literature review to provide justification for your study.
The term “evaluation” was used in the search (line 151). It should be added to line 148.

Results:
Please make sure that the description and numbers of articles described in the Study Selection section aligns with Figure 1.

Other suggestions:
Title: Shouldn’t only the first alphabet of the first word be capitalized?
WHO Health-Promoting Schools Framework and WHO health-promoting components were mentioned in the article. If they present the same concept, I suggest authors to maintain consistency throughout the article.
Table 1. Please list studies reviewed in chronological order; use the correct format for Table 1 title.
Several grammar errors were identified. Please correct them throughout the manuscript. For example, line 305.
Citations in the references should follow the guideline and format provided by the journal.

---

## Round 0.2 · accepted · Accept

Thank you for the manuscript update and the detailed answer. There are no more critical remarks.

·

Basic reporting

No comment

Experimental design

No comment

Validity of the findings

No comment